# Revealing Abnormal Functional Coupling in Schizophrenia Using PhaseICA, a Complex-Valued ICA Framework

Liang Ma

*Tri-Institutional Georgia State University/Georgia Institute of Technology/Emory University Center for Translational Research in Neuroimaging and Data Science*
Atlanta, Georgia, USA
lma12@gsu.edu

Tulay Adali

*Department of Computer Science and Electrical Engineering University of Maryland, Baltimore County*
Baltimore, Maryland, USA
adali@umbc.edu

Masoud Seraji

*Tri-Institutional Georgia State University/Georgia Institute of Technology/Emory University Center for Translational Research in Neuroimaging and Data Science*
Atlanta, Georgia, USA
mseraji1@gsu.edu

Vince D. Calhoun

*Department of Computer Science Georgia State University*
Atlanta, Georgia, USA
vcalhoun@gsu.edu

*Abstract*—Schizophrenia (SZ) is characterized by disruptions in functional coupling across brain regions, often linked to neurotransmitter dysregulation and abnormal synaptic plasticity. While traditional functional connectivity analyses focus on pairwise, synchronous interactions, they often overlook complex, temporally delayed coupling patterns that may underlie SZ pathology. In this study, we applied PhaseICA, a complex-valued independent component analysis method, to resting-state fMRI data to investigate spatiotemporal brain dynamics in SZ. The PhaseICA method captures both stationary and nonstationary brain waveforms by decomposing whole-brain fMRI signals into spatially independent analytic components, incorporating phase delays via the Hilbert transform and entropy-bound minimization. This framework enables identification of temporally offset wave patterns without relying on predefined regions or templates. Our results revealed significantly altered amplitudes of both stationary and travelling brain waves in individuals with SZ, with some abnormalities correlating with cognitive scores. These findings highlight spatiotemporally delayed functional interactions as a core feature of SZ and underscore the utility of PhaseICA in uncovering clinically relevant brain dynamics beyond conventional connectivity models.

*Index Terms*—Schizophrenia, functional coupling, PhaseICA, amplitude, phase delay

## I. INTRODUCTION

Schizophrenia (SZ) is a severe psychiatric disorder characterized by disturbances in cognitive, perceptual, and emotional processing. A growing body of evidence suggests that abnormal functional coupling—particularly atypical patterns of connectivity—plays a central role in the pathophysiology of schizophrenia [1, 2]. Many theories attribute dysfunctional coupling to aberrant synaptic plasticity and neurotransmitter dysregulation, especially involving dopaminergic and glutamatergic systems [3, 4, 5], as well as gamma-aminobutyric acid (GABA) interneuron dysfunction [6]. A deeper understanding of dysfunctional coupling may help elucidate the underlying neural mechanisms and guide the development of targeted interventions for schizophrenia.

Abnormal functional coupling in schizophrenia is typically studied using static or dynamic functional connectivity between pairs of brain regions based on predefined templates [7, 8]. These analyses primarily focus on interactions between spatially fixed regions or networks. While most approaches assess pairwise functional coupling, it is increasingly recognized that multi-region or network interactions may provide a more comprehensive view of brain dysfunction. Some studies have expanded beyond pairwise analyses using multivariate metrics, such as total correlation, to evaluate connectivity among multiple regions [9]. However, these methods become computationally expensive as the number of regions increases. Moreover, most current techniques assume synchronous (i.e., non-delayed) co-activation across regions, thereby neglecting temporal delays in interregional communication—a phenomenon increasingly reported in recent studies [10].

Crucially, spatially distinct networks may exhibit similar fluctuation profiles with temporal offsets. Abnormal functional coupling driven by neurotransmitter dysregulation may involve delayed interregional effects, which are often overlooked by conventional analyses. This weakness is derived from the inability of detection on travelling waves, because the space and time in the travelling wave equation $y = A(x)A(t)cos(\phi(x) + \phi(t))$ could not be linear separated in the real-domain.

In this study, we applied a phase-integrated approach, PhaseICA [11], to investigate SZ brain dynamics in complex domain. This method decomposes spatially independent brain

waves by first transforming spatiotemporal fMRI signals into analytic waveforms using the Hilbert transform, followed by complex entropy bound minimization to optimize the spatially independence between components. The wave component not only reflects stationary activities within brain networks, but also reflect nonstationary brain activities. The wave pattern represents a recurring activities of functional coupling across all brain voxels, capturing spatial distributions of temporal delays without being constrained by the number of regions or reliance on pre-defined templates. Due to the simultaneous analysis of delayed functional coupling across the entire brain, this method thereby reduces the risk of false positives associated with conventional high-dimensional connectivity analyses. Our results reveal significantly abnormal amplitudes in the stationary and nonstationary wave on SZ patients, and some of them is significantly related with cognition scores.

## II. DATASET DESCRIPTION

The Functional Imaging Biomedical Informatics Research Network (FBIRN) Phase III release is a large, harmonized multi-site neuroimaging cohort comprising 362 adults, including 186 healthy controls (HCs) and 176 patients diagnosed with schizophrenia or schizoaffective disorder. Participants were recruited from seven U.S. academic medical centers, utilizing six Siemens Tim-Trio scanners and one GE Discovery MR750 scanner. All participants underwent structured clinical interviews based on the SCID-DSM-IV-TR and were confirmed to be clinically stable for at least two months prior to scanning.

Imaging acquisition followed the standardized FBIRN "traveling-phantom" protocol to reduce cross-scanner variability. Resting-state functional MRI (rsfMRI) data were acquired using a harmonized T2*-weighted gradient-echo echo-planar imaging (EPI) protocol applied uniformly across all 3T scanners without modification. Each rsfMRI scan consisted of 162 volumes, corresponding to a duration of 5 minutes and 24 seconds. Acquisition parameters were as follows: repetition time (TR) = 2000 ms, echo time (TE) = 30 ms, voxel size = $3.4 \times 3.4 \times 4$ mm, field of view (FOV) = $220 \times 220$ mm, and flip angle = 77°. During scanning, participants were instructed to keep their eyes closed but remain awake.

Following quality control procedures, 166 SZ patients and 161 HC participants were retained for analysis. The dataset was subsequently divided into two subsets: a discovery cohort (83 SZs and 81 HCs) and a validation cohort (83 SZs and 80 HCs). No significant differences were observed in age or gender distributions between the discovery and validation datasets, nor between SZ and HC groups within each dataset (all $p > 0.5$).

## III. METHOD

We identified shared reliable brain waves in the HC and SZ groups using a group-based analysis pipeline similar to our previous NeuroMark framework [17]. The main difference is that we employed the PhaseICA method [11] to reflect the phase delay information at the same time. The phaseICA

is a complex-valued independent component analysis (ICA) approach. Unlike conventional ICA methods that identify only spatially stationary brain waves (analogous to static intrinsic networks), this method is designed to capture both spatially stationary and nonstationary waves, which propagate across brain regions with delayed phases. The identification of reliable brain waves across the population was performed in below steps: 1) Hilbert transformation and complex whitening, 2) complex entropy bound minimization (CEBM) optimization, 3)selection of reliable brain Waves and 4) Back-reconstruction of wave time series.

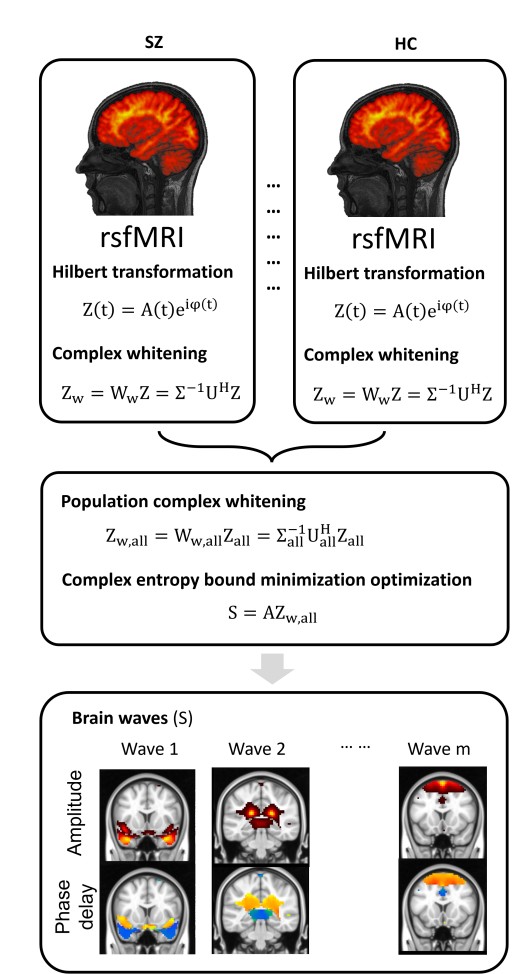

Fig. 1. The pipeline to generate the brain waves across population.

### A. Hilbert Transformation and Complex Whitening

For each participant, let $Y \in \mathbb{R}^{T \times V}$ be the rsfMRI signals of this participant, which includes $V$ oscillated variables ($y_v, v = 1, \ldots, V$) with continuous observations at $T$ time points. First of all, Brain signal of each subject is normalized by the general mean $\mu$ and standard deviation $\sigma$ of all voxels with $Y = (Y - \mu)/\sigma$. The mean value of each spatial unit $y_v$ is removed by $\tilde{y}_v(t) = y_v(t) - \frac{1}{T} \sum_{t=1}^{T} y_v(t)$ to ensure that correlations and covariances are computed correctly.

For each spatial location, the real-valued time series is transformed into a complex analytic signal:

$$z_v(t) = y_v(t) + i \cdot \mathcal{H}\{y_v(t)\}, \quad \forall v = 1, 2, \ldots, V \quad (1)$$

where $\mathcal{H}\{y_v(t)\}$ denotes the Hilbert transform of $y(t)$, defined as:

$$\mathcal{H}\{y_v(t)\} = \text{P.V.} \int_{-\infty}^{\infty} \frac{y_v(\tau)}{t - \tau}\, d\tau \quad (2)$$

Here, P.V. stands for the Cauchy principal value of the integral, which is a special type of integral used when a traditional (definite) integral does not exist due to a singularity—a point where the function becomes infinite or undefined. The analytic signal $z_v(t) = Ae^{i\theta}$,, which combines the original signal and its Hilbert transform, allows us to compute instantaneous amplitude, phase, and frequency from complex time series.

Stacking across $V$ spatial units, we obtain the complex-valued matrix $Z \in \mathbb{C}^{V \times T}$. Top $K$ eigenvectors and eigenvalues of the complex matrix $\tilde{Z}$ are obtained from fast SVD decomposition, which is approximately estimated using a probabilistic singular value decomposition (SVD) method [12], expressed as:

$$\tilde{Z} = U \Sigma V^H, \quad (3)$$

where $U \in \mathbb{C}^{T \times K}$ is the matrix of left singular vectors, $\Sigma \in \mathbb{R}^{K \times K}$ is a diagonal matrix of singular values, and $V^H \in \mathbb{C}^{K \times V}$ is the Hermitian transpose of the right singular vectors. Here, $K \leq V$ denotes the number of reduced components. Finally, analytic signal $\tilde{Z}$ is whitened by whitening matrix $W_w = \Sigma^{-1} U^H$ and the whitened matrix $\tilde{Z}_w$ are expressed as :

$$\tilde{Z}_w = W_w \tilde{Z} = \Sigma^{-1} U^H (U \Sigma V^H) = V^H \quad (4)$$

Thus, whitening actually maps the data to the space spanned by the right singular vectors, and the result has unit variance. The whitened signals from all subjects ($P = 1, \ldots, p$) are then concatenated, $\tilde{Z}_{all} = [\tilde{Z}_{w1}, \tilde{Z}_{w2}, \ldots, \tilde{Z}_{wp}]$ The concatenated matrix is reduced to $K$ components using singular value decomposition (SVD) as $\tilde{Z}_{all} = U_{all} \Sigma_{all} V_{all}^H$. The reduced whitened population signals $\tilde{Z}_{w,all} = V_{all}^H$ are obtained from the Hermitian transpose of the right singular vectors. Here, $V_{all}^H \in \mathbb{C}^{K \times S}$ represents the reduced complex-valued signal matrix across the population.

### B. Complex Entropy Bound Minimization (CEBM) Optimization

The whitened data $\tilde{Z}_w$ includes $K$ spatial components $\tilde{z}_{w,i}$, $i = 1, \ldots, K$. We seek independent components defined as:

$$S = W_d \tilde{Z}_w, \quad (5)$$

where $S \in \mathbb{C}^{K \times T}$ contains the estimated sources and $W_d$ is the demixing matrix. The goal is to find an optimal $W_d$ such that the derived sources are as independent as possible. A natural cost function is mutual information minimization, which leads to the following objective:

$$\min_{W_d} J(W_d) = \sum_{i=1}^{K} H(s_i) - \log|\det(W_d)| \quad (6)$$

where $H(\cdot)$ denotes the differential entropy, and its computation requires estimation of the source density function. Typical ICA methods make use of fixed nonlinearities, and their extensions to the complex domain assume circularity (e.g., Complex FastICA [14]), but this assumption is violated by noncircular complex signals derived from Hilbert transformations. In this study, we adopt the Complex Entropy Bound Minimization (CEBM) method [15] to optimize the independence of complex-valued sources. CEBM minimizes mutual information as in our objective ( 6) and makes use of the maximum entropy principle to approximate the density using measuring functions. The approach introduces two entropy bounds: For a linear decomposition $[s_1, s_2] = B[u, v]$, the first entropy bound for each sources $s_k$ could be described as:

$$H_1(s_k) \leq \log(|\det(B)|) + H(u) + H(v). \quad (7)$$

For a polar decomposition $[s_1, s_2] = Br[\sin\theta, \cos\theta]$, and $u$ and $v$ are the complex magnitude and the principal value of the argument of $u + iv$, respectively, the second entropy bound could be described as:

$$H_2(s_k) \leq \log|\det(B)| + E[\log(r)] + H(r) + \log(2\pi). \quad (8)$$

It computes both bounds and chooses the tighter one:

$$\hat{H}(s_k) = \min\{H_1(s_k), H_2(s_k)\}. \quad (9)$$

The entropy bound $\hat{H}(s_k)$ serves as a surrogate for the true entropy $H(s_k)$. This framework minimizes a tight upper bound of entropy, thus avoiding direct estimation of the probability density function. At each iteration $t$, the algorithm updates each row $w_i$ of the demixing matrix $W_d$ separately, using conjugate gradient descent to minimize $\hat{H}(s_k)$ with the iterative function 10, 11, 12:

$$u_n^{(t)} = \frac{\partial J(w_n^{(t)})}{\partial \overline{w}_n} - \text{Re}\left( w_n^{(t)H} \frac{\partial J(w_n^{(t)})}{\partial \overline{w}_n} \right) w_n^{(t)} \quad (10)$$

$$w_n^{(t+1)} = w_n^{(t)} - \mu \frac{u_n^{(t)}}{\left\| u_n^{(t)} \right\|} \quad (11)$$

$$w_n^{(t+1)} = \frac{w_n^{(t+1)}}{\left\| w_n^{(t+1)} \right\|} \quad (12)$$

where $\mu$ is the step size and $\overline{w}_n$ is the conjugate of $w$. The $\frac{\partial J(w_n^{(t)})}{\partial \overline{w}_n}$ is the conjugate gradient of $J(w_n)$. This optimization also removes the need to constrain $W_d$ to be orthogonal, allowing for a more flexible search space. Detailed computation could be referred to the [15] article.

Each complex component $S_k = a_k + ib_k$, where $a_k = \text{Re}(S_k)$, $b_k = \text{Im}(S_k)$, represents a spatiotemporal brain wave of the form: $S_k = A_k \cos(\theta_k)$ The real and imaginary parts correspond to spatial modes of the wave at phase $\theta = 0$ and $\theta = \frac{\pi}{2}$, respectively. The magnitude $A_k$ describes the amplitude envelope, and its spatial distribution is calculated as the norm of the complex value:

$$A_k = |S_k| = \sqrt{a_k^2 + b_k^2} \quad (13)$$

The spatial distribution of phase is given by:

$$\theta_k = \arg(S_k) = \arctan\left(\frac{b_k}{a_k}\right) \quad (14)$$

### C. Selection of Reliable Brain Waves

We selected reliable brain waves based on a metric, defined as the maximum rotated similarity between corresponding components extracted from two independent datasets. Specifically, the similarity between two complex-valued components $u$ and $v$ was computed as the maximum Hermitian inner product after rotating one component by a phase shift $\theta \in [-\pi, \pi]$:

$$r(u, v) = \max_{\theta \in [-\pi, \pi]} \frac{|\langle u, v e^{i\theta}\rangle|}{\|u\| \cdot \|v e^{i\theta}\|} \quad (15)$$

Here, $\|\cdot\|$ denotes the Euclidean norm of a complex vector, and $\langle \cdot, \cdot \rangle$ represents the Hermitian inner product. Brain waves were considered reliable if their reproducibility exceeded a threshold of 0.6.

### D. Back-Reconstruction of Wave Time Series

The spatial pattern of each brain wave component was first normalized by dividing by its maximum absolute amplitude, such that the maximum spatial amplitude of each wave equals 1. The corresponding time series for the waves, denoted by the complex-valued mixing matrix $\mathbf{M} = [m_1, m_2, ..., m_k] \in \mathbb{C}^{T \times K}$, was back-reconstructed using the Moore-Penrose pseudoinverse of the spatial matrix $\mathbf{S}$:

$$\mathbf{M} = \mathbf{X}\mathbf{S}^\dagger \quad (16)$$

Here, $\mathbf{X} \in \mathbb{C}^{T \times V}$ is the original spatiotemporal signal matrix, and $\mathbf{S}^\dagger$ is the Moore-Penrose pseudoinverse of $\mathbf{S} \in \mathbb{C}^{K \times V}$, the spatial patterns of the $K$ brain waves.

For each wave $k$, the mean amplitude across time points was computed as:

$$\bar{A}_k = \frac{1}{T}\sum_{t=1}^{T} \|m_k(t)\| \quad (17)$$

Group-level differences in the mean wave amplitude $\bar{A}_k$ were statistically tested between the SZ and HC groups.

## IV. RESULTS

We applied group-level brain wave decomposition with 100 components to both the discovery and validation datasets, each comprising SZ and HC participants with no significant differences in age or gender distributions. After excluding components unrelated to gray matter, we identified 67 reliable brain waves based on reproducibility criteria. We then examined the mean amplitude of each wave's time series over the entire duration. Among the 67 reliable brain waves, seven exhibited statistically significant differences in mean amplitude between the SZ and HC groups (FDR-corrected $p < 0.001$). The spatial distributions of these six waves are presented in Figure 2, and the corresponding results from two-sample $t$-tests are shown in Figure 3.

These differences in wave amplitude suggest abnormal functional coupling across multiple brain regions in SZ patients. The following waves demonstrated notable group-level alterations:

- **Wave C14:** SZ patients show the decreased wave amplitude in the medial part of the primary motor cortex (M1), related motor functions.
- **Wave C25:** This wave shows the peak region in the superior occipital gyrus, involved in the anti-phase coupling between posterior medial cortex(PCC) and the tail of the caudate nucleus. The SZ patients show significant decreased wave amplitude.
- **Wave C26:** SZ patients showed increased coupling among the ventricles, thalamus, and striatum, with ventricular activity exhibiting anti-phase coupling relative to thalamic and striatal activity.
- **Wave C35:** SZ patients show the decreased anti-phase coupling between superior occipital gyrus and ventral occipital lobe.
- **Wave C42:** The decreased wave activities in the bilateral insular regions in SZ patients.
- **Wave C45:** Elevated coupling was observed among the insula, superior temporal gyrus (STG), and cerebellum in SZ patients. Notably, insular activity was in anti-phase with STG and cerebellar activity.
- **Wave C46:** This wave primarily engaged the visual cortex in the occipital lobe and revealed reduced coupling or modularity between the primary and secondary visual cortex in SZ patients.
- **Wave C48:** The wave shows the anti-phase coupling between medial region and lateral region in occipital lobe. SZ patients show reduced coupling.
- **Wave C49:** Increased coupling strength was observed between the dorsomedial prefrontal cortex (dmPFC) and ventromedial prefrontal cortex (vmPFC), with an anti-phase relationship characterizing SZ patients.
- **Wave C52:** SZ patients show the decreased amplitude in the lateral occipital lobe.

Generally, SZ patients show a decreased wave amplitude in the region related with primary functions (e.g. related visual and motor functions), including C14, C35, C46, C48 and C52. On the contrary, SZ patients has a increased amplitude for waves (C26, C45 and C49), which spatial regions associated with many high-level functions. We do not see the association of wave amplitudes with both PANSS positive and negative scores. But We have found their significance association (FDR corrected p¡0.01) with 5 congition scores (Speed Of Processing, Attention Vigilance, Working Memory, Visual Learning and CMINDS composite). Specifically, the "Speed Of Processing" score is related with C25 and C45 amplitudes. The "attention vigilance" score is significantly and positively related (r=0.27) with C25 amplitude but negatively related (r=-0.21 and -0.22) with C26 and C49 wave amplitudes. The "Working Memory" score is related (r=0.25) with C25 amplitude. "Visual Learning" ability is associated (r=-0.24)

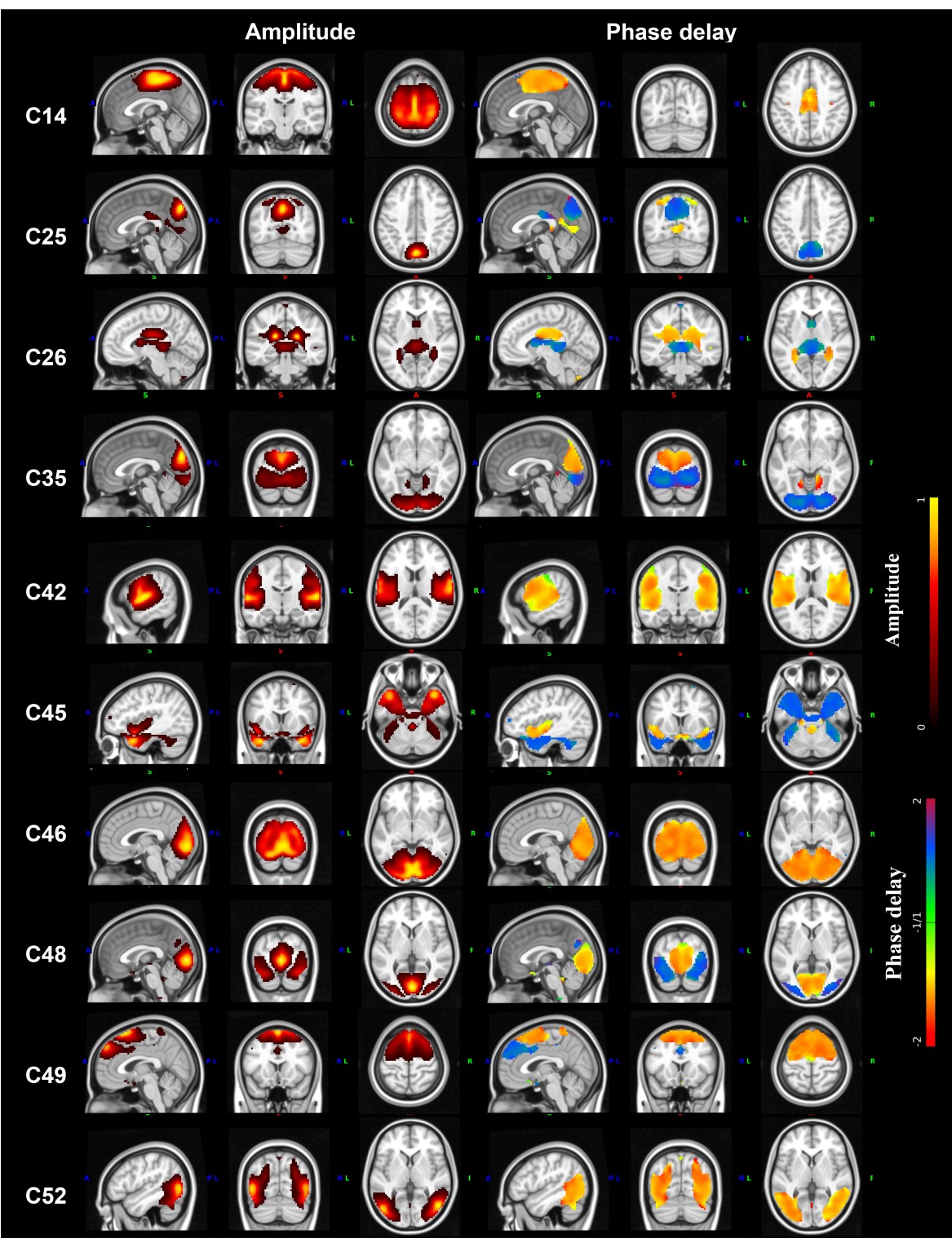

Fig. 2. The amplitude and phase delay distributions of significant brain waves. **Left three columns:** Amplitude distribution (Sagittal, coronal and Axial views). **Right three columns:** Phase delay distribution (Sagittal, coronal and Axial views). Both maps are masked to include only regions with wave amplitude greater than 0.1.

with C45 amplitude, while the "CMINDS composite" score is related (r=0.24 and -0.25) with C25 and C45 amplitudes.

More importantly, the brain regions involved in Waves C45, C46, C49 include many cortical areas known to be enriched with neurotransmitter receptors vulnerable in schizophrenia, as reported by Hansen et al. [16]. This convergence suggests a strong association between altered brain wave dynamics and underlying neurotransmitter dysregulation in SZ.

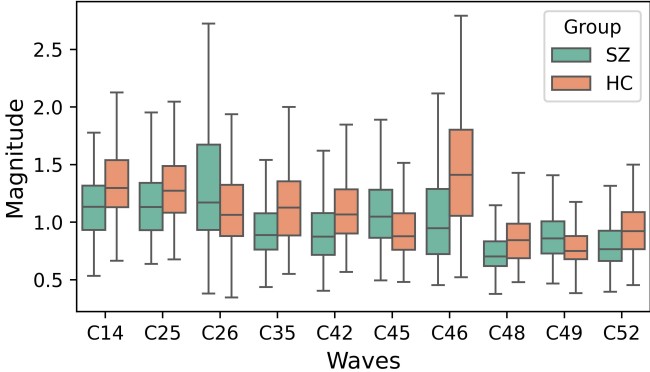

Fig. 3. Amplitude comparison between healthy controls (HC) and schizophrenia (SZ) patients in significant brain waves, using two-sample $t$-tests across the full dataset. Each box represents the mean amplitude of a wave time series in each group. The significance of all above components is Benjamini-Hochberg FDR-corrected, with the p value below 0.001.

## V. CONCLUSION AND DISCUSSION

In this study, we applied PhaseICA, a complex-valued ICA framework, to uncover abnormal spatiotemporal brain waves in schizophrenia. Unlike conventional static connectivity or pairwise dynamic methods, PhaseICA captures multi-region interactions with inherent phase delays, offering a biologically grounded characterization of functional coupling. Our analysis revealed significant alterations in wave amplitudes among SZ patients, particularly in circuits involving the thalamus, striatum, insula, and visual and motor cortices. These abnormalities co-localized with cortical distributions of neurotransmitter receptors known to be affected in schizophrenia, supporting the hypothesis that altered wave dynamics may reflect underlying neuromodulatory dysfunction. In the future, it would be better to refinement of spatial components using subject-specific optimization process to get an accurate evaluation on amplitude. The hypothesis on the reflection of neurotransmitters should be validated by the medication effect, as well as symptom changes. Furthermore, the phase information to reflect the movement of brain activities should be further analyzed via a dynamic framework with functional directionality. Overall, PhaseICA opens a new direction for investigating large-scale brain dynamics in mental disorders and may inform future strategies for biomarker development and neuromodulation-based interventions.

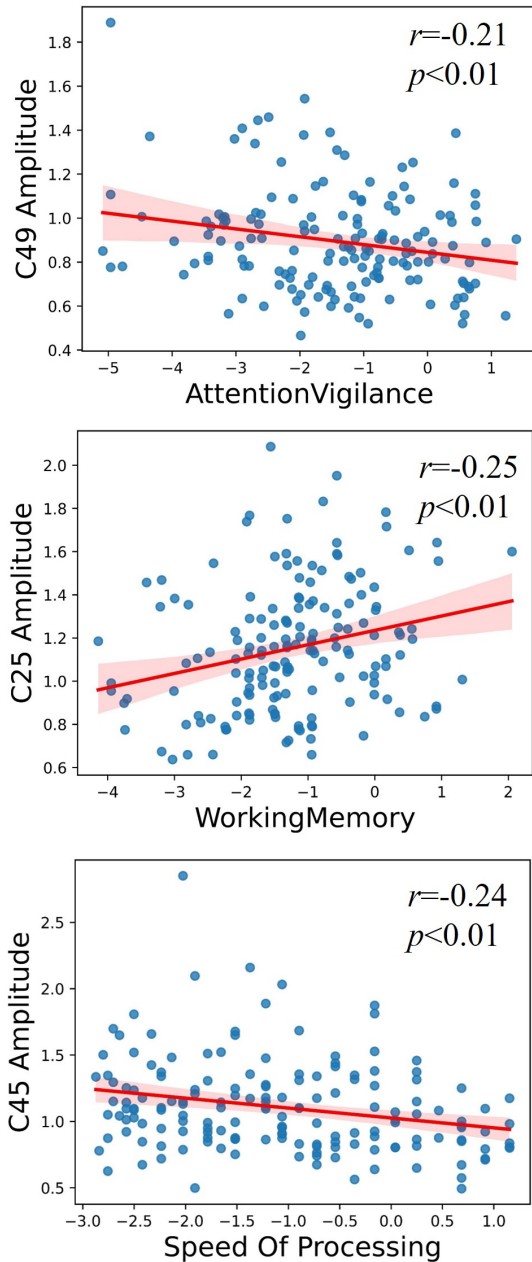

Fig. 4. Three examples on the association between abnormal amplitudes and SZ cognition scores with significance Benjamini-Hochberg FDR-corrected p below 0.01.

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
