# OpenReview forum: "Abnormal functional coupling strength revealed by brain waves in schizophrenia patients"
_IEEE.org/EMBS/BHI/2025/Conference — BHI 2025_

### Official Review · Reviewer_V8dX · 2025-06-30
**Reproducibility concern due to complex processing pipeline**

**Confidence:** 2
**Clarity Of Writing:** good
**Clinical Significance:** good
**Methodological Novelty:** fair
**Overall Rating:** 5

**Experiments And Results:**

good

**Questions For The Authors:**

1.	Please follow the IEEE format for all references.
2.	Too much use of ‘novel’ instead of implying how the proposed method is new. For instance, in the fourth paragraph of the introduction, PhaseICA is introduced. However, the reasoning behind selecting the Hilbert transform and entropy-bound minimization is absent.
3.	In the introduction section, a discussion on state-of-the-art research would strengthen the establishment of the need and significance of this research. Currently, the research gap is not clear that the authors are trying to address.
4.	Could you please add further explanations of Figure 2? While each row represents a brain wave, what is in the different columns? I presume all images are for SZ patients. Wouldn’t it be nice to include a few discussions and evidence on HC, as you included a comparison of HC and SZ in Figure 3?
5.	Authors are encouraged to discuss the reproducibility of their work, and if feasible, include a link to source code.

**Strengths:**

PhaseICA incorporates both amplitude and phase information in brain connectivity. Identified wave patterns support biological plausibility as they correspond to brain regions and neurotransmitter systems known to be involved in schizophrenia.

**Summary Of The Paper:**

The paper presents PhaseICA, analyzing spatiotemporal brain dynamics in schizophrenia using resting-state fMRI data. It captures both amplitude and phase information to extract brain waves that reveal temporally delayed and spatially organized functional interactions.

**Weaknesses:**

The methodological complexity and use of complex-valued signals may limit accessibility and reproducibility. Would it be possible to link the abnormal wave patterns to specific clinical symptoms or severity in SZ?

---

### Official Review · Reviewer_WaUt · 2025-07-16
**Abnormal functional coupling strength revealed by brain waves in schizophrenia patients**

**Confidence:** 4
**Clarity Of Writing:** good
**Clinical Significance:** great
**Methodological Novelty:** good
**Overall Rating:** 7

**Experiments And Results:**

good

**Questions For The Authors:**

Points for improvement in the document:
-	The legend of Figure 2 should be revised: The labels on the right and left sides appear to be reversed.
-	It could be outlined potential future steps to build on these findings. What strategies or modifications are envisioned to further improve the outcomes presented?
-	A period should be added at the end of the sentence on the second line of page 3, as well as on the second line following Equation 12.

**Strengths:**

An advantage of PhaseICA is its ability to analyse delayed brain connectivity on a whole-brain scale while reducing false positives. The method also identified altered brain wave amplitudes in schizophrenia patients, matching regions linked to neurotransmitter receptor vulnerabilities.

**Summary Of The Paper:**

This study presents PhaseICA, a new method to analyse brain dynamics in schizophrenia. It transforms fMRI signals into analytic waveforms using the Hilbert transform, then extracts spatially independent brain patterns by minimizing complex entropy. The resulting brain waves reveal recurring connectivity patterns across the brain, without relying on predefined templates or region limits.

**Weaknesses:**

Figure 3: Which factors or techniques could contribute to minimizing measurement variability, as reflected in the error bars?

---

### Official Review · Reviewer_Bndn · 2025-07-16
**This review commends the study for introducing PhaseICA, a novel complex‑valued ICA method that uncovers previously hidden, time‑lagged “brain‑wave” networks and validates its findings in a large multi‑site schizophrenia cohort. It highlights the work’s methodological innovation, strong sample size, and mechanistic links to neurotransmitter systems, but notes limitations such as short scan duration, cross‑sectional design, and incomplete reporting on phase metrics and clinical correlations. Overall, the review concludes that PhaseICA offers promising insights into spatiotemporal dysconnectivity in schizophrenia while calling for longitudinal analyses, expanded clinical validation, and greater transparency to facilitate replication.**

**Confidence:** 5
**Clarity Of Writing:** excellent
**Clinical Significance:** excellent
**Methodological Novelty:** excellent
**Overall Rating:** 8

**Experiments And Results:**

excellent

**Questions For The Authors:**

How sensitive are PhaseICA results to preprocessing choices (e.g., motion scrubbing or global signal regression)?

Did you examine correlations between altered wave amplitudes and clinical symptoms, cognitive scores, or antipsychotic dose?

PhaseICA detects both amplitude and phase information—why were group differences reported only for amplitude?

Could the method differentiate schizophrenia from other psychiatric disorders (e.g., bipolar disorder) in future studies?

What computational resources and run‑times are required, and is PhaseICA code publicly available for replication?

**Strengths:**

Methodological innovation: PhaseICA models time‑lagged, multi‑region coupling that conventional ICA or static functional‑connectivity methods miss, offering richer physiological interpretability.

Robust sample and validation: A well‑powered, harmonized multi‑site dataset was split into discovery and validation cohorts with balanced demographics, enhancing generalizability.

Reliability controls: Only waves with reproducibility > 0.6 were analyzed, limiting false positives.

Biological convergence: Abnormal waves align with regions enriched for vulnerable neurotransmitter receptors, adding mechanistic plausibility.

**Summary Of The Paper:**

The authors present PhaseICA, a complex‑valued independent‐component framework that decomposes resting‑state fMRI signals into “brain waves” capturing both amplitude and phase‑delay information among distributed regions. Using the large multi‑site FBIRN Phase III cohort (166 schizophrenia patients, 161 healthy controls), they identified 67 reproducible waves and found that seven showed significant amplitude differences between groups. Altered waves involved the thalamus, striatum, insula, prefrontal, motor and visual cortices, and their spatial patterns overlapped cortical areas rich in neurotransmitter receptors implicated in schizophrenia pathology. The study concludes that PhaseICA reveals previously hidden spatiotemporal disconnectivity and could inform biomarker development for psychiatric disorders.

**Weaknesses:**

Without longitudinal or treatment data, it is unclear whether the observed dysconnectivity tracks symptom changes or medication effects (patients were simply “clinically stable” for ≥2 months). Also, the analysis emphasizes mean amplitude differences; phase‑delay metrics, clinical correlations, and potential confounds (e.g., head motion, medication dose) are not reported. Those, these may be reported in a future manuscript.

---

### Official Review · Reviewer_urdL · 2025-07-17
**Abnormal functional coupling strength revealed by brain waves in schizophrenia patients**

**Confidence:** 4
**Clarity Of Writing:** great
**Clinical Significance:** fair
**Methodological Novelty:** good
**Overall Rating:** 4
**Final Rating:** 7

**Experiments And Results:**

great

**Questions For The Authors:**

1. How does PhaseICA perform compared to standard ICA or graph-based methods?

2. Were multiple comparison corrections (e.g., Bonferroni) applied when testing across 67 waves?

3. Have the authors considered validating PhaseICA with EEG or MEG?

**Strengths:**

1. The introduction of PhaseICA, a complex-valued ICA framework that captures both amplitude and phase delay. It allows for the detection of temporally delayed and spatially distributed brain coupling patterns.

2. Figures are clear and concise.

**Summary Of The Paper:**

The study introduces PhaseICA, a novel complex-valued independent component analysis method, to investigate abnormal functional coupling in schizophrenia (SZ). Unlike traditional approaches that assess static or pairwise connectivity, PhaseICA captures spatiotemporal brain waves that reflect both amplitude and phase delay of brain activity across regions. Using resting-state fMRI data from the FBIRN Phase III dataset, the authors applied PhaseICA to extract 67 reproducible brain waves. They identified six waves with significantly different amplitudes between SZ patients and healthy controls.

**Weaknesses:**

1. Although the study reports statistically significant differences in brain wave amplitudes between schizophrenia patients and healthy controls, it does not explore their clinical relevance. Without associating these abnormalities with clinical symptoms, functional outcomes, or treatment response, the translational value remains limited.

2. The paper does not quantitatively compare PhaseICA with established approaches like standard ICA or graph-based connectivity analyses.

3. The analysis involves comparisons across 67 brain waves, but it is not clearly stated whether multiple comparison correction was applied. For this many tests, correction methods such as Bonferroni are typically required.

4. Given the study’s focus on phase delay and oscillatory dynamics, a comparison or validation with higher-temporal-resolution modalities like EEG would be valuable.